# Tubulin Polyglutamylation by TTLL1 and TTLL7 Regulate Glutamate Concentration in the Mice Brain

**DOI:** 10.3390/biom13050784

**Published:** 2023-05-01

**Authors:** Yashuang Ping, Kenji Ohata, Kenji Kikushima, Takumi Sakamoto, Ariful Islam, Lili Xu, Hengsen Zhang, Bin Chen, Jing Yan, Fumihiro Eto, Chiho Nakane, Keizo Takao, Tsuyoshi Miyakawa, Katsuya Kabashima, Miho Watanabe, Tomoaki Kahyo, Ikuko Yao, Atsuo Fukuda, Koji Ikegami, Yoshiyuki Konishi, Mitsutoshi Setou

**Affiliations:** 1Department of Cellular and Molecular Anatomy, Hamamatsu University School of Medicine, 1-20-1 Handayama, Higashi-ku, Hamamatsu, Shizuoka 431-3192, Japan; pingyashuang1989@gmail.com (Y.P.); ohata.mail@gmail.com (K.O.); largo.kenji@gmail.com (K.K.); t.sakamoto0731@gmail.com (T.S.); ariful222222@gmail.com (A.I.); lilixu2412@gmail.com (L.X.); hengsen19110@gmail.com (H.Z.); chenbin19101@gmail.com (B.C.); yanjing20105@gmail.com (J.Y.); etofmhr@gmail.com (F.E.); n.chiho830@gmail.com (C.N.); kahyo@hama-med.ac.jp (T.K.); yaoik@kwansei.ac.jp (I.Y.);; 2Department of Gastroenterology, Graduate School of Medicine, The University of Tokyo, 7-3-1 Hongo, Bunkyo-ku, Tokyo 113-0033, Japan; 3International Mass Imaging Center, Hamamatsu University School of Medicine, Hamamatsu, 1-20-1 Handayama, Higashi-ku, Hamamatsu, Shizuoka 431-3192, Japan; 4Department of Behavioral Physiology, Faculty of Medicine, University of Toyama, 2630 Sugitani, Toyama-shi, Toyama 930-0194, Japan; takao@cts.u-toyama.ac.jp; 5Genetic Engineering and Functional Genomics Unit, Frontier Technology Center, Graduate School of Medicine, Kyoto University, Kyoto 606-8501, Japan; miyakawa@fujita-hu.ac.jp; 6Institute for Comprehensive Medical Science Division of Systems Medicine, Fujita Health University, Aichi 470-1192, Japan; 7Department of Neurophysiology, Hamamatsu University School of Medicine, Hamamatsu, 1-20-1 Handayama, Higashi-ku, Hamamatsu, Shizuoka 431-3192, Japan; mihow@hama-med.ac.jp (M.W.);; 8Department of Biomedical Sciences, School of Biological and Environmental Sciences, Kwansei Gakuin University, 1 Gakuen Uegahara, Sanda, Hyogo 669-1330, Japan; 9Department of Anatomy and Developmental Biology, Graduate School of Biomedical and Health Sciences, Hiroshima University, 1-2-3 Kasumi, Hiroshima 734-8553, Japan; 10Department of Applied Chemistry and Biotechnology, University of Fukui, 3-9-1 Bunkyo, Fukui-shi, Fukui 910-8507, Japan; 11Department of Systems Molecular Anatomy, Institute for Medical Photonics Research, Preeminent Medical Photonics Education & Research Center, 1-20-1 Handayama, Higashi-ku, Hamamatsu, Shizuoka 431-3192, Japan

**Keywords:** glutamate, TTLL, polyglutamylation, tubulin, post-translational modification, MALDI IMS

## Abstract

As an important neurotransmitter, glutamate acts in over 90% of excitatory synapses in the human brain. Its metabolic pathway is complicated, and the glutamate pool in neurons has not been fully elucidated. Tubulin polyglutamylation in the brain is mainly mediated by two tubulin tyrosine ligase-like (TTLL) proteins, TTLL1 and TTLL7, which have been indicated to be important for neuronal polarity. In this study, we constructed pure lines of *Ttll1* and *Ttll7* knockout mice. *Ttll* knockout mice showed several abnormal behaviors. Matrix-assisted laser desorption/ionization (MALDI) Imaging mass spectrometry (IMS) analyses of these brains showed increases in glutamate, suggesting that tubulin polyglutamylation by these TTLLs acts as a pool of glutamate in neurons and modulates some other amino acids related to glutamate.

## 1. Introduction

Glutamate is the most abundant excitatory neurotransmitter in the brain and is at the crossroad of multiple metabolic pathways; its cellular homeostasis is essential for normal brain function, such as motor learning and recognition memory [1,2,3]. Previous studies have suggested that glucose in the brain acts as a small glutamate pool, and about 20% of released glutamate in the brain is derived from glucose [4,5]. However, the way that glutamate is synthesized in the brain still remained unclear. Extracellular concentration of glutamate in the brain is regulated to be low (less than 1.0 µmol/L) compared to the intracellular concentration (about 10.0 mmol/L) [6]. Increased extracellular concentration of glutamate causes neurotoxic effects, which lead to several neurodegenerative diseases such as Alzheimer’s disease and Huntington’s disease [3,7]. Glutamate in neurons is synthesized from glutamine released from adjacent astrocytes, and the glutamine is metabolized in astrocytes from glutamate taken up by excitatory amino acid transporters (EAATs) on the astrocytes [8]. The glutamate–glutamine cycle is important for neuronal activities, but it has not been clear where glutamate in neurons is stored.

Tubulin in neurons undergoes various post-translational modifications (PTMs). Polyglutamylation is one of the most abundant PTMs in the brain, where a polymer of different numbers of glutamates is added at the C-terminal tail of α- and β-tubulins [9,10,11]. Polyglutamylation of α- and β-tubulins have distinct effects and is done by specific tubulin tyrosine ligase-like (TTLL) proteins [11,12,13]. TTLL1 and TTLL7 play major roles in catalyzing polyglutamylation in the brain [11,12]. These site-specific polyglutamylations change the interaction with microtubule-associated proteins (MAPs) and motor proteins, which generate neural polarity [14,15,16]. 

Matrix-assisted laser desorption/ionization (MALDI) Imaging mass spectrometry (IMS) can directly acquire and visualize the position of various molecules based on their *m/z* [17,18]. It is widely used in the detection of metabolites, drugs, lipids, peptides, and proteins [19]. Due to the poor ionization efficiency of amino acids, they cannot be directly detected by MALDI IMS [20]. Therefore, to detect glutamate and other amino acids in the brain of *Ttll* knockout mice, we used the on-tissue chemical derivatization method with 2,4-diphenyl-pyranylium tetrafluoroborate (DPP-TFB) [20,21]. 

In this study, we constructed pure lines of *Ttll1* and *Ttll7* knockout mice to explore the effects of tubulin polyglutamylation on mice. *Ttll1* and *Ttll7* knockout decreased polyglutamylated α- and β-tubulins, respectively, which change the electric properties of tubulin. MALDI IMS analyses of these brains showed increase in not only glutamate but also γ-aminobutyric acid (GABA) and some other amino acids related to glutamate. These results suggest that tubulin polyglutamylation by these TTLLs acts as a pool of glutamate in neurons, and some other amino acids were modulated by the glutamate as well. As morphological concerns, *Ttll1/Ttll7* double knockout reduced the number of neurofilaments in the apical dendrites of cortical neurons, resulting in the reduction of the diameter of dendritic shafts. 

## 2. Materials and Methods

### 2.1. Generation of Knockout Mice

*Ttll1* knockout mice were constructed as described in our previous research [22]. To generate *Ttll7* knockout mice, 129/Sv ES cell clone, in which the *Ttll7* allele including exon 2–3 (1989 bp from AfeI site) was replaced by a TKneo cassette, was injected in C57BL/6 blastocysts. The heterozygous mice generated from the germline chimera were continuously crossed to C57BL/6J to reduce the effects of backcross. Genotyping of mice was performed by PCR by using two kinds of forward primers; 5′-GAATTCACTGGCACTCACTCAAATG-3′ and 5′-CCGGTCTTGTCGATCAGGATG (TKneo cassette) and a reverse primer; 5′-AAACACAGCACTTGCAGAGGGTT-3′. Animals were treated according to the guideline of the Hamamatsu University School of Medicine.

### 2.2. Antibodies

The primary antibodies for immunostaining were GT335 (Gift from B. Edde and Dr. C. Janke, 1:2000), Neurofilament M (RMO14.9; Cell Signaling Technology, 1:25), and Map2 (ab5622; Merck Millipore, 1:1000). The secondary antibodies were Alexa fluor 488 (Thermo Fisher Scientific, 1:1000) and Alexa fluor 568 (Thermo Fisher Scientific, 1:1000). The antibodies used for immunoblot were TTLL1 (Ikegami et al., 2010, 1:1000), TTLL7 (Ikegami et al., 2006, 1:1000), GT335 (1:20,000), α-tubulin (DM1A; Sigma-Aldrich, 1:50,000), β-tubulin (TUB2.1; Sigma-Aldrich, 1:2000), tyrosinated tubulin (1A2; Sigma-Aldrich, 1:2000), acetylated tubulin (6-11B-1; Sigma-Aldrich, 1:20,000), neurofilament 200 (NE14; Sigma-Aldrich, 1:500), neurofilament 160 (NN18; Sigma-Aldrich, 1:250), glyceraldehyde-3-phosphate dehydrogenase (GAPDH, 6C5; Sigma-Aldrich, 1:5000), CLIP170 (Sigma-Aldrich, 1:500), Map1A (HM-1; Sigma-Aldrich, 1:1000), Map1B (Sigma-Aldrich), 1:1000; Map2 (HM-2; Sigma-Aldrich), 1:1000; Tau-1 (MAB3420; Merck Millipore, 1:10,000), dynein (74.1; Merck Millipore, 1:1000), KIF1A (no. 16; BD Biosciences, 1:500), KIF5 (H2; Merck Millipore, 1:500), and KIF17 (Sigma-Aldrich, 1:1000). The antibodies used for 2D electrophoresis were GT335 (1:20,000), α-tubulin (DM1A; 1:4000), β-tubulin (TUB2.1; 1:4000).

### 2.3. Chemicals

Scopolamine methyl bromide and sodium hydroxide were purchased from Wako Pure Chemical Industries (Osaka, Japan). Pilocarpine hydrochloride was purchased from the Tokyo chemical industry. Glutamate standard, DPP-TFB, and 2,5-dihydroxybenzoic acid were purchased from Sigma-Aldrich. 1 mol/L ammonium formate solution was purchased by Kanto Chemical Co., Inc. (Tokyo, Japan). LC/MS grade methanol, ultrapure water, acetonitrile, formic acid, and isopropanol were purchased from Wako Pure Chemical Industries (Osaka, Japan).

### 2.4. Mouse Behavior Analysis

Mice were maintained under a 12:12-h light–dark cycle at 22 ± 2 °C, with a relative humidity of 40% to 60%, and with access to chow and water ad libitum. All animal care and procedures performed in this study were undertaken according to the rules and regulations of the Guide for the Care and Use of Laboratory Animals. The experimental protocols were approved by the Animal Care Committee of Laboratory Animals of the Kyoto University (approved number: MedKyo 08165, MedKyo 09539) and the Hamamatsu University School of Medicine. The raw data of the behavioral tests and the information about each mouse are accessible on the public database “Mouse Phenotype Database” (http://www.mouse-phenotype.org/; accessed on 1 May 2023).

### 2.5. Immunoblot

To prepare brain lysate, mouse brain tissues were homogenized in RIPA buffer (25 mM Tris-HCl, pH 7.6, 150 mM NaCl, 1% NP-40, 1% sodium deoxycholate, 0.1% SDS). The supernatants were collected after centrifuging at 14,000 rpm for 10 min at 4 °C. Microubule fraction was prepared as described previously [13]. Protein concentrations were measured by the Bradford protein assay (Bio-Rad). Equal amounts of proteins were separated by SDS-PAGE electrophoresis and transferred onto a polyvinylidene fluoride membrane (Merck Millipore). After incubations of primary and secondary antibodies, the immunoreactivity of the membrane was detected using an ECL kit (GE Healthcare).

### 2.6. 2D Electrophoresis

We followed the methods described previously [13]. Tissues were homogenized in lysis buffer (7 M urea, 2 M thiourea, 2% CHAPS/40 mM DTT, 2% IPG buffer pH 3–10), and samples were subjected to isoelectric focusing (IEF) in an immobilized pH linear gradient gel (IPG gel) of pH 4.5–5.5, 24 cm in length (GE Healthcare). Second-dimensional gel electrophoresis was performed using laboratory cast SDS/polyacrylamide gels (8%).

### 2.7. Wire Hang Test

The mouse was placed on the wire mesh and then slowly reversed so that the mouse could hold it tightly and would not fall off. The latency of the fall was recorded with a cut-off time of 60 s.

### 2.8. Grip Strength Test

Grip strength was conducted as previously described [23]; a grip strength meter was used to assess forelimb grip strength. Mice were lifted and held by their tail so that their forepaws could grasp a wire grid. The mice were then gently pulled backward by the tail until they released the grid. The peak force applied by the forelimbs of the mouse was recorded in Newtons (N). Each mouse was tested 3 times and the largest value was used for statistical analysis.

### 2.9. Porsolt Forced Swim Test

Porsolt forced swim test was conducted as previously described [24]. The test device consists of four plastic cylinders (20 cm high × 10 cm diameter), and the cylinders were filled by water (21–23 °C) up to a height of 7.5 cm. Mice were placed into the cylinder, and both immobility and the distance traveled were recorded over a 10 min test period. Data acquisition and analysis were performed automatically using Image TS software.

### 2.10. Social Interaction in Home Cage

Social interaction monitoring in the home cage was conducted as previously describe, by a system that automatically analyzes social behavior in the home cages of mice [25]. One *Ttll7* knockout and one wild type (WT) mouse were placed together in a home cage. Their social behavior was then monitored for 1 day. Outputs video analysis was performed by Images HA software. Social interaction was measured by counting the number of particles detected in each frame; two particles indicated that the mice were not in contact with each other, and one particle demonstrated contact between the two mice.

### 2.11. Elevated plus Maze

An elevated plus maze test was conducted as previously described [24]. The elevated plus maze consisted of two open arms (30 × 5 cm) and two enclosed arms of the same size with 15 cm high transparent walls, and the arms were connected by a central square (5 × 5 cm). The number of entries into the open and enclosed arms and the time spent in the open or enclosed arms were recorded during a 10 min test period. The percentage of entries into open arms, time spent in open arms(s), number of total entries, and total distance traveled (cm) were calculated. Data acquisition and analysis were performed automatically using Image EP software.

### 2.12. Kindling Test

Continuous injection of pilocarpine was performed to induce the kindling model [26,27]. The mice were placed in an observation chamber for about 30 min for habituation. A total of 4 injections were required: first, subcutaneous injection of 1 mg/kg of scopolamine methyl bromide; 30 min later, intraperitoneal injection of 100 mg/kg of pilocarpine hydrochloride and commencement of the observation of mice behaviors; 60 min later, intraperitoneal injection of 100 mg/kg of pilocarpine hydrochloride; 75 min later, intraperitoneal injection of 20 mg/kg of pilocarpine hydrochloride; then, continue to observe for 90 min. Seizure Score: 0: normal behavior, no abnormality. 1: immobilization, lying on belly. 2: head nodding, facial, forelimb, or hindlimb myoclonus. 3: continuous whole-body myoclonus, myoclonic jerks, tail held up stiffly. 4: rearing, tonic seizure, falling down on its side. 5: tonic-clonic seizure, falling down on its back, wild rushing and jumping. 6: death. The seizure score is counted every 5 min, and the highest score is scored.

### 2.13. MALDI IMS

WT and *Ttll* knockout mice were euthanized by cervical dislocation. Subsequently, the brain tissues were rapidly removed and frozen in dry ice powder. The frozen brain tissues were cut at a thickness of 10 µm using a cryostat microtome (Leica CM1950, Leica Microsystems). The sagittal brain sections are about 1 mm in the laterolateral axis with respect to the bregma, and the coronal brain sections are about −1.2 mm in the interaural axis with respect to the bregma. The brain sections were thawed and mounted on indium tin oxide-coated slides (ITO glass, Matsunami Glass Ind., Ltd., Osaka, Japan) at −20 °C. Transferred the frozen sections to a vacuum pump (Diaphragm Type Dry Vacuum Pump DTC-21, Ulvac Kiko, Inc., Miyazaki, Japan) and left therein for 20 min to keep brain sections from condensation, the sections warm to room temperature during drying. About 200 µL or 100 µL of derivatization solution containing DPP-TFB (Sigma-Aldrich, St. Louis, MO, USA; Cat. #R246875) in methanol (1.33 mg/mL), was sprayed manually to each sagittal or coronal section. After that, an automatic sprayer (TM-Sprayer, HTX Technologies) was used to spray 40 mg/mL of 2,5-dihydroxybenzoic acid in 50% methanol onto the samples. For different pH of glutamate standards, 1 mg/mL glutamate standard solution was prepared and the same volume of different concentrations of formic acid solution and sodium hydroxide solution were added to the glutamate standard solution to get a pH range: 2.08, 3.81, 6.17, 9.06, 10.08, 11.71. Then, 50 μL of different pH glutamate standard solutions were mixed with 50 μL of 1.33 mg/mL DPP-TFB (in methanol) and 50 μL of 40 mg/mL 2,5-dihydroxybenzoic acid (in 50% methanol). A total of 2 μL of the mixed solution was added onto indium tin oxide-coated slides and dried in the vacuum pump at room temperature. The acidic condition of WT and *Ttll7* knockout mice brain sections was prepared by adding 3% formic acid to all solutions. MALDI IMS was performed using Fourier-transform ion cyclotron resonance mass spectrometer (Solarix XR, Bruker Daltonics). Our laser focus was set to “medium” for sagittal sections and “small” for coronal sections, both laser pulses were 1000 Hz, and the number of irradiations was 200 shots for sagittal sections and 50 shots for coronal sections. The raster step size was set to 120 µm for sagittal sections and 30 µm for coronal sections. Positive ions in a mass range of *m/z* 210–510 for sagittal sections and 247.25–600 for coronal sections were obtained. 

### 2.14. Immunohistochemistry

Phosphate Buffered Saline and 4% paraformaldehyde were perfused transcardially into the mice. Tissues were embedded in paraffin. For immunohistochemical examination, 4 µm thick sections are deparaffinized and rehydrated. Antigen retrieval was performed with 10 mM citrate buffer pH 6.0 for 40 min at 96 °C with a pressure cooker. Sections were blocked in 5% goat serum for 30 min at room temperature. After blocking, sections were incubated with primary antibodies overnight at 4 °C, followed by a 1 h incubation of secondary antibodies at room temperature. Finally, fluorescence was detected under a confocal microscope (FV-1000, Olympus, Central Valley, PA, USA).

### 2.15. Transmission Electron Microscopy

Anesthetized mice were transcardially perfused with fixation solution (30 mM Hepes-OH buffer (pH 7.4), 2% paraformaldehyde, and 2.5% glutaraldehyde) pre-warmed at 37 °C. The brain was cut with a razor blade to obtain coronal sections (about 1 mm), then, the small piece of the region that contained the somatosensory cortex was soaked in the fixation solution for 2 h at 4 °C. Tissue samples were washed 4 times with 60 mM Hepes-OH (pH 7.4) and left overnight at 4 °C. Samples were treated with osmium tetroxide, dehydrated, and embedded in resin. The region corresponding to layers 2–3 of the somatosensory cortex was confirmed by observing the semithin section, and the ultrathin section (60-nm to 80-nm) was prepared. Samples were observed by JEM-1220 electron microscope (JEOL, Tokoyo, Japan) at 8000 kV.

### 2.16. LC-MS/MS 

The section was prepared in the same way as MALDI IMS and a total of 4 methods were used for LC-MS/MS. In the first method, 150 µm coronal sections were homogenized with extraction solution (50 µL/mg tissue), 80% acetonitrile solution containing 0.2% formic acid, and 10 ng/mL internal standard of glutamate, then centrifuged at 20,000× *g* for 10 min at 4 °C. The supernatant was collected and stored at −20 °C. The samples were diluted and evaporated to dryness before LC-MS/MS, then reconstituted in 60% acetonitrile and filtered with a 0.2 µm filter. The standard curve was constructed by diluting the glutamate standard with 60% acetonitrile, and the concentration range was 10–250 ng/mL, then, the same amounts of internal standards were added to all standard solutions. LC-MS/MS was conducted using an Acquity UPLC system (Waters, Milford, MA, USA), an autosampler, and a 4000 QTRAP linear ion trap quadrupole mass spectrometry system (Sciex, Foster City, CA, USA). All of the procedures were controlled by Analyst software (version 1.6.1, Sciex). Chromatographic separations were used Acquity UPLC BEH Amide column (2.1 × 150 mm with 1.7 μm particle size). The mobile phase consisted of A (0.1% formic acid) and B (acetonitrile) at a flow rate of 0.2 mL/min. The injection volume was 5 µL. The gradient elution was applied as follows: 0–1 min, 1% A; 1–7 min, 1–70% A; 7–9 min, 70% A; 9–9.1 min, 70–1% A; 9.1–15 min, 1% A. The mass spectrometric detection was performed in positive mode with an electrospray ionization (ESI) source. The second method changed the extraction solution to acetonitrile-isopropanol-aqueous (3:3:2, *v*/*v*/*v*) [28], changed the internal standard of glutamate to 50 ng/mL, changed the mobile phase B to acetonitrile containing 0.1% formic acid, and changed the gradient elution to: 0–1 min, 20% A; 1–7 min, 20–80% A; 7–9 min, 80% A; 9–9.1 min, 80–20% A; 9.1–15 min, 2% A. Other procedures are the same as the first method. The third method, 150 μm sagittal brain section was used for the experiment. The position was similar to the MALDI IMS sample. Before adding the extraction solution, add 2 mg/mL DPP-TFB (30 μL/mg brain tissue), vortex, and then reactions for 30 min, dry up the samples, other procedures the same as the second method. The mass spectrometric of 4000 QTRAP MS/MS system detection was performed using an electrospray ionization (ESI) source in positive mode. The following settings were optimized for achieving the highest signal intensity: ion spray voltage: +/−5500 V, source temperature: 600 °C, curtain gas: 40, gas-1: 40, gas-2: 70, Q1/Q3 resolution: unit, and interface heater: on. The multiple reaction monitoring (MRM) transitions for each analyte and IS, as well as their respective optimum MS parameters, including declustering potential (DP), entrance potential (EP), collision energy (CE), and cell exit potential (CXP), are shown in Appendix A. Next, we used method 4 to detect the glutamate. An amount of 250 μm whole brain coronal sections were used for experiments, and the brain sections were homogenized with 1.89% formic acid in ultrapure water containing 50 ng/mL internal standard of glutamate [29]. After homogenization and centrifuging, 100 μL supernatant was added to 400 μL ACN with 1% formic acid for protein precipitation and centrifuged at 20,000× *g* for 10 min at 4 °C. The supernatant was filtered with a 0.2 μm filter, then evaporated and dissolved in 0.2% formic acid in water/ACN (95/5, *v*/*v*). The calibration glutamate standard was prepared in 0.2% formic acid in water/ACN (95/5, *v*/*v*) containing the internal standard of glutamate, and the concentration ranges are 10–250 ng/mL, LC-MS/MS was conducted by a Waters triple quadrupole mass spectrometer. All of the procedures were controlled by MassLynx software (version V4. 1). Separations were used Intrada amino acid column (100 mm × 3 mm and 3 μm particle size). Mobile phase A was 0.3% formic acid in ACN. Mobile phase B was ACN/100mM ammonium formate in water = 20/80). The flow rate was 0.5mL/min and the gradient elution was applied as follows: 0–4 min, 80% A; 4–4 min, 0% A; 14–6 min, 0% A; 16–6.1 min, 0–80% A; 16.1–20 min, 80% A. The autosampler temperature was maintained at 10 °C with an aliquot of 5 μL of the prepared sample injected into the system. The mass spectrometer was equipped with an electrospray ionization (ESI) source, and samples were analyzed in positive mode. The glutamate standards transitioned from *m/z* 148.14 to 84.10, and the internal standard glutamate transitioned from *m/z* 154.2 to 89.10. We optimized the TQ-MS parameters for maximum sensitivity as follows: scan mode: multiple reaction monitoring (MRM), capillary voltage: 4.0 kV, cone voltage: 22 V, glutamate standard collision energy: 18.0 V, internal standard glutamate collision energy: 12.0 V, source temperature of 150 °C, desolvation temperature: 500 °C, desolvation gas flow: 600 L/h, cone gas flow: 150 L/h, N2 gas pressure: 7.0 bar.

## 3. Results

### 3.1. Generation of Ttll7 Knockout Mice, Ttll1 and Ttll7 Knockout Reduced Polyglutamylated α- and β-Tubulins, Respectively

In this study, we freshly constructed pure lines of *Ttll1* and *Ttll7* knockout mice (Figure 1A) by repeated backcrosses using WT mice of C57BL/6 background. We confirmed that these *Ttll1* and *Ttll7* knockout mice did not express TTLL1 and TTLL7 proteins, respectively (Figure 1B). Brain lysates from these mice showed a significant reduction in polyglutamylated α- and β-tubulins by the knockouts of *Ttll1* and *Ttll7,* respectively (Figure 1B, E- α/βTub), whereas total amounts of both α- and β-tubulins (Figure 1B, αTub and βTub) or other PTMs, tyrosination, and acetylation, did not change (Figure 1B, Tyr-Tub, Ac-Tub, respectively).

### 3.2. Ttll1 and Ttll7 Knockout Changed the Tubulin Electrical Properties

The addition of polyglutamate chains on the C-terminal tail of tubulin increases the negative charge of the protein, which could change the affinity to other proteins. In order to examine the electric properties, we performed high-resolution two-dimensional electrophoresis, followed by immunoblots with antibodies to polyglutamylated tubulins or tubulin itself. *Ttll1* knockout strongly reduced the polyglutamylation signal of α-tubulins (Figure 1C, left panels, 1KO), while *Ttll7* knockout reduced the signal of polyglutamylated β-tubulins (Figure 1C, left panels, 7KO). In the same manner, *Ttll1/Ttll7* double knockout decreased signals from both polyglutamylated α- and β-tubulins (Figure 1C, left panels, dKO), however, the total amount of both tubulins was almost unchanged (Figure 1C, middle panels). Lateral migration patterns of each gel showed that the loss of TTLL1 and TTLL7 reduced the amount of the polyglutamylated tubulins with a highly negative charge (Figure 1C, right panels), implicating the loss of glutamate chains bound to each tubulin. These results suggest that TTLL1 and TTLL7 were responsible for the polyglutamylation of α- and β-tubulins, respectively, but did not regulate the expression of both tubulins in the brain.

### 3.3. Ttll Knockout Mice Showed Several Abnormal Behaviors

A battery of behavioral tests was used to elucidate the behavioral effects of *Ttll* knockout, and *Ttll* knockout mice showed several abnormal behaviors. In general health and neurological screen, *Ttll7* knockout mice were significantly heavier (WT 33.19 ± 2.25, 7KO 34.96 ± 2.40, *t*_(38)_ = −2.40, *p* = 0.02, unpaired t test, Figure 2A), and the latency to fall off the wire mesh was significantly shorter (WT 60.00 ± 0.00, 7KO 47.65 ± 14.63, *t*_(38)_ = 3.77, *p* = 0.0005, unpaired t test, Figure 2C), when we compared the two results, the shorter latency in the wire hang test may be related to heavier body weight. There were no significant differences between the *Ttll7* knockout and WT mice in body temperature and grip strength (WT 36.36 ± 0.74, 7KO 36.19 ± 0.81, *t*
_(38)_ = 0.71, *p* = 0.48, unpaired t test, Figure 2B, WT 0.88 ± 0.24, 7KO 0.86 ± 0.18, *t*
_(38)_ = 0.30, *p* = 0.77, unpaired t test, Figure 2D). 

In the Porsolt forced swim test, *Ttll7* knockout showed normal immobility during the entire 10 min on both day 1 and day 2, but in the first 5 min block of day 1, *Ttll7* knockout mice exhibited a significantly increased immobility posture compared with WT mice. On both day 1 and day 2, *Ttll7* knockout mice activity was higher in the last 3 min block (Figure 2E). The distance traveled was not significantly different for the entire 10 min on both day 1 and day 2, but in the first 5 min block of day 1, *Ttll7* knockout mice exhibited a significantly shorter distance traveled compared with WT mice (Figure 2F). These results suggest that the *Ttll7* knockout mice showed an increased depression-like phenotype during the initial period.

In the social interaction test in the home cage, the *Ttll7* knockout mice showed an increased mean number of particles during the night. This suggests that *Ttll7* knockout mice showed more locomotor activity during the night (Figure 2G). In the elevated plus maze test, there was no significant difference between *Ttll7* knockout and WT mice in the distance traveled. During the first 5 min block of the test, the *Ttll7* knockout mice distance traveled was significantly greater (Figure 2H). The result suggests that the *Ttll7* knockout mice were hyperactive in the initial period.

Continuous injection of pilocarpine hydrochloride was used to induce the chemical kindling [26,27]. In 135 min of observation, *Ttll1*(Het)/*Ttll7*(Homo) mice had a significantly longer duration of obvious epilepsy-like symptoms (highest score) than WT mice (WT 3.23 ± 0.15, 7KO 3.75 ± 0.10, *t* _(4)_ = 5.17, *p* = 0.0066, unpaired t test, Figure 2I). The result suggests that loss of *Ttll1* and *Ttll7* genes was associated with the antiseizure function or prevention of epilepsy.

### 3.4. Knockout of Ttll1 and Ttll7 Increased Glutamate Concentration in Mouse Brain

The concentration of glutamate in the brain is tightly controlled, and any defects disturbing this balance can cause neuronal disorders such as seizures and loss of motor coordination [30]. We examined if these *Ttll* knockout affect glutamate concentration in the brain by using MALDI IMS. MALDI IMS directly acquires and visualizes the position of various molecules based on their *m/z*. Amino acids can be detected by on-tissue chemical derivatization methods with DPP-TFB, as we reported [21] (Appendix A). We detected glutamate in the sagittal brain section of *Ttll1/Ttll7* double knockout and WT mice. The signal intensity of glutamate was increased in *Ttll1/Ttll7* double knockout mice compared to WT mice brains, especially in the cortex (CTX), hippocampal formation (HPF), and cerebellum (CB) regions (Figure 3A). From the result, we can consider that loss of tubulin polyglutamylation increases the amount of free glutamate in the brains of *Ttll* knockout mice. Interestingly, significant changes in other amino acids were also found. The amount of GABA, another neurotransmitter synthesized from glutamate in the neurons, was increased not only in the hypothalamus and midbrain, but also in regions similar to glutamate (Figure 3A). Concentrations of extracellular GABA, as well as glutamate, are tightly controlled; extracellular GABA is mainly collected by astrocytes, converted into glutamine, and recycled to GABA again in neurons. Distributions of glutamine, valine, and leucine/isoleucine were also changed upon *Ttll* knockout; these amino acids are also derived from glutamate (Figure 3A). We exported the intensity of each spot and drew the line graphs of relative intensity frequency, we can see the relative intensity of glutamine in HPF and CB regions slightly increased in *Ttll1/Ttll7* double knockout mice brain, other regions of amino acids were obviously increased (Appendix A). Subsequently, we statistically analyzed it and calculated the effect size, all selected regions of those amino acids have significant differences except for the HPF region of glutamine. Glutamate and valine in all selected regions and leucine/isoleucine in the HPF region had a large effect size. GABA in all selected regions, glutamine in CB, leucine/isoleucine in CTX, and CB regions had moderate effect size. Glutamine in CTX and HPF regions had a small effect size (Table 1).

Then, we focused on the cortical region and selected the coronal cortical region of *Ttll* knockout and WT mice brain for detection. Those amino acids were found to have similar changes to the sagittal cortical region (Figure 3B). The line graphs were used to compare the relative intensity frequency between *Ttll* knockout and WT mice, and we can see that all of the amino acids in *Ttll* knockout mice brain have higher relative intensity (Figure 3C). Then, we statically analyzed the intensity of each spot and calculated the effect size, those amino acids have significant differences and large effect size between *Ttll* knockout and WT mice (Table 2). These findings indicate that tubulin polyglutamylation acts as a glutamate pool in neurons and modulates some other amino acids. 

### 3.5. Ttll Knockout Does Not Change the Overall Structures of the Mouse Brain

We observed the macroscopic morphology and coronal histological structure of WT and *Ttll1/Ttll7* double knockout mice, and there were no obvious differences between them (Figure 4A). The brain weight was not affected by the *Ttll* knockout (Figure 4B). We used glutamylated tubulins antibody and anti-MAP2 antibody reaction with coronal section, the polyglutamylated tubulins were mainly located in the surface cortical region (Figure 4C). Then, we investigated the effect of polyglutamylase deficits on the cortical structure. In *Ttll* knockout mice, neither thickness of the cerebral cortex nor neuron intensity was significantly affected. (Figure 4D–F). The *Ttll* knockout did not affect basic brain structures. 

### 3.6. Ttll1/Ttll7 Double Knockout Mice Decreased Neurofilaments in the Dendrites

Glutamate concentration was greatly changed, especially in the cortex region of *Ttll* knockout mice, so we examined the detailed structure in this region by immunohistochemistry. Signals of polyglutamylated tubulins in the layer I were markedly reduced in *Ttll1* knockout mice (Figure 5A, 1KO indicated by asterisks), whereas most of the signals remained in *Ttll7* knockout mice (Figure 5A, 7KO). In the apical dendrites of cortical neurons (layer II/III), signals from glutamylated tubulin were abolished only by *Ttll1/Ttll7* double knockout (Figure 5A, dKO indicated by arrowheads) as well as in the layer I; however, they remained in the single knockout mice. These results indicate that cortically projecting axons in the layer I were enriched in tubulin that received TTLL1-dependent polyglutamylation mainly on α-tubulin. On the other hand, both TTLL1 and TTLL7 might act compensatory in apical dendrites (layer II/III). The abolished polyglutamated tubulin signal in the cortex of *Ttll1/Ttll7* double knockout mice is where the increased glutamate signal was detected. Interestingly, neurofilaments in *Ttll1/Ttll7* double knockout mice were remarkably reduced in the apical dendrites at layer II/III (ANOVA, F_(3, 10)_ = 6.55, followed by Dunnett’s test, *p* = 0.0053, Figure 5B), whereas immunoblot analysis indicates that the expression level of neurofilament was not affected (ANOVA, F _(3, 33)_= 4.43, followed by Dunnett’s test, *p* = 0.021, Figure 5C). The neurofilament distribution in dendrites seems to be in accordance with the localization of polyglutamylated tubulin in dendrites. Observation by an electron microscopy also showed the reduction of neurofilaments in the apical dendrites at layer II/III of *Ttll1/Ttll7* double knockout mice (Appendix A). We confirmed that the reduction of neurofilaments in dendrites was not caused by the change in the length (Appendix A). The thicknesses of axonal dendritic shafts in *Ttll1/Ttll7* double knockout mice were significantly smaller compared with those in WT mice (Appendix A; ANOVA, F_(3, 8)_ = 4.06, followed by Dunnett’s test, *p*= 0.026, Appendix A). It was surprising that polyglutamylation on tubulin mainly affected the neurofilament distribution, specifically in the dendrite, while it had little effect on the morphology of tubulin, which is the direct substrate of TTLLs. 

We have shown that tubulin polyglutamylation changes the motility of some motor proteins [22]. In search of proteins that respond to the tubulin de-polyglutamylation, we performed the tubulin co-sedimentation assay using the brain from each knockout mouse (Figure 5D). Several microtubule-associated proteins and motor proteins were reduced in tubulin fraction by loss of TTLL1 and TTLL7. In particular, CLIP170, MAP1A, dynein, and KIF1A were strongly reduced by the *Ttll1/Ttll7* double knockout.

### 3.7. LC-MS/MS Quantitative Analyses of Glutamate Indicated no Difference between the Brains of Ttll Knockout and WT Mice

For the Liquid chromatography-coupled tandem mass spectrometry (LC-MS/MS) measurement, four methods were used to detect glutamate in mice brains. Glutamate in the concentration range of 10–250 ng/mL was used to draw the standard curve (Figure 6A). In the first method, LC-MS/MS was conducted by a 4000 QTRAP linear ion trap quadrupole mass spectrometry system. Three groups of *Ttll* knockout and WT mice half brain coronal sections were used to detect glutamate with the acidic extraction solution, as we have reported previously [21]. There was no significant difference between *Ttll* knockout and WT mice (Figure 6B). In the second method, we changed the extraction solution to neutralized, acetonitrile-isopropanol-aqueous (3:3:2, *v*/*v*/*v*) [28], mobile phase B changed to acetonitrile containing 0.1% formic acid, and also changed the gradient elution. There was no significant difference between *Ttll7* knockout and WT mice (WT 2395.33 ± 337.58, 7KO 2297.00 ± 173.18, *t*_(4)_ = 0.45, *p* = 0.68, unpaired t test, Figure 6C). In the third method, we used DPP-TFB derivatization glutamate. We calculated the derivatization glutamate (DPP-Glu) area and derivatization internal standard glutamate (DPP-IS-Glu) area ratio, there was no significant difference between *Ttll7* knockout and WT mice (WT 2.46 ± 0.53, 7KO 2.36 ± 0.2, *t*_(4)_ = 0.22, *p* = 0.83, unpaired t test, Figure 6D). In the fourth method, LC-MS/MS was conducted by the triple quadrupole mass spectrometer, the glutamate extraction by 1.89% formic acid in water containing the internal standard of glutamate [29]. The amount of glutamate extracted from the whole brain coronal section of *Ttll7* knockout and WT mice was not significantly different (WT 3718.33 ± 111.33, 7KO 3837.92 ± 545.10, *t*_(10)_ = 0.53, *p* = 0.61, unpaired t test, Figure 6E). The LC-MS/MS and MALDI IMS results are inconsistent.

## 4. Discussion

In this study, we freshly generated pure-line *Ttll1* and *Ttll7* knockout mice. *Ttll1* and *Ttll7* knockout reduced polyglutamylated α- and β-tubulins, respectively. These results are in accordance with previous findings that TTLL1 acts mainly on the α-tubulin [11], and TTLL7 acts specifically on β-tubulin in cultured neuronal cells [12]. The *Ttll7* knockout showed several abnormal behaviors, suggesting that loss of the *Ttll7* gene is associated with anti-epileptic function and plays an important role in basis of mood and locomotor activity. During brain development, polyglutamylation on the α-tubulin retains at a high level, and that on the β-tubulin gradually increases [31,32], supporting the assertion that the polyglutamylation on tubulin in the brain is important for brain maturation. Intriguingly, *Ttll1/Ttll7* double knockout mice showed a reduction of not only polyglutamylated α- and β-tubulins, but also neurofilaments in the apical dendrites of cortical neurons, resulting in the reduction of the diameter of dendritic shafts, which is in accordance with previous reports that the loss or disruption of neurofilaments could decrease dendritic diameter [33,34]. Polyglutamylation of tubulin changes the electric condition on the tubulin surface, which could affect the interactions of MAPs and motor proteins to the microtubules. We confirmed that the binding affinities of some MAPs and motor proteins were changed by the *Ttll* knockout. CLIP170, MAP1A, dynein, and KIF1A were strongly reduced by the *Ttll1/Ttll7* double knockout. It has been reported that neurofilaments are transported by tubulin motors [35,36,37,38]. Summarizing these findings, we speculate that polyglutamylation on tubulin mediated by TTLL1 and TTLL7 changes the binding affinity of several MAPs and motor proteins, then modulates neurofilament arrangement in the dendrites. 

MALDI IMS analysis of the brains from these *Ttll* knockout mice showed a significant increase in glutamate concentration, *Ttll1/Ttll7* double knockout increased the most, and the *Ttll7* knockout was higher than *Ttll1* knockout, but the immunohistochemistry result showed that *Ttll1* knockout reduced more polyglutamylated tubulin signal in cortex regions (Figure 5A). First, this may be due to the different reactions of the antibody to α- and β-tubulins during immunostaining. The GT335 antibody we used is a monoclonal antibody that can detect both α- and β-tubulins branch points of glutamate chains, which reacts strongly with α-tubulin and weakly with β-tubulin [39]. Second, there are many polyglutamylated tubulin isotypes in the adult mammalian brain that add different numbers of glutamates; previous studies have demonstrated that the α1/2 isotype has four additional glutamates, α4 has five, βI has three, βII has six, βIII has four, and βIV has five [40,41]. β-tubulin isotype has more glutamate residues than the α-tubulin isotypes, so *Ttll7* knockout mice release more glutamate than *Ttll1* knockout.

Previous studies have demonstrated that extracellular glutamate increases during seizures [42]. *Ttll* knockout mice had a longer duration of epilepsy-like symptoms in the kindling test, suggesting that *Ttll* knockout mice have released more glutamate to extracellular space, which is consistent with our MALDI IMS results. Glutamate acts as the most common excitatory neurotransmitter in the brain, while excess glutamate causes neurotoxic effects both in acute neurological diseases such as cerebral ischemia, brain injury, epilepsy, and hepatic encephalopathy, and in chronic neurological diseases such as Alzheimer’s disease, amyotrophic lateral sclerosis (ALS), Huntington’s disease, and glaucoma [43]. Besides glutamate, concentrations of many other amino acids also showed remarkable increases by the *Ttll* knockout. GABA is the main inhibitory neurotransmitter, which is synthesized from glutamate by glutamate decarboxylases (GADs) in neurons [44]. In the brain, it is recycled by the glutamate (GABA)-glutamine cycle between neurons and astrocytes [45], so it was reasonable that the amount of GABA and glutamine increases with glutamate. Previous studies have shown that physiological concentrations of GABA can inactivate GAD through feedback regulation [46], so when *Ttll* knockout increased the glutamate concentration in neurons, only a small amount of glutamate was synthesized to GABA. Therefore, *Ttll* knockout mice showed more excitability in the kindling test. Branched-chain amino acids (BCAA) are important nitrogen donors for glutamate synthesis in the brain [47]. Recently, the “glutamate-BCAA cycle” has been proposed; this mechanism not only supports the synthesis of glutamate, it also provides a complementary mechanism that supplements the action of the glutamate-glutamine cycle in the instantaneous “disposing” of synaptic glutamate [48]. Increased concentrations of glutamate can cause excitotoxicity in neurons, while the cytosolic carboxypeptidase 1 (CCP1) can catalyze deglutamylation to promote neuron survival [49]. Therefore, when *Ttll* knockout increased the glutamate, the increase of valine and leucine/isoleucine could support the idea of “disposing” of over-released glutamate in *Ttll* knockout mice brains. We speculate that polyglutamylation on tubulin in neurons could act as a glutamate pool in neurons and modulate some other amino acids.

It is surprising that when we use LC-MS/MS to quantitatively analyze the glutamate in the mouse brain, there is no obvious difference between WT and *Ttll* knockout mice, which is why we used four LC-MS/MS methods. LC-MS/MS did not correspond to the results obtained by MALDI IMS. This may be due to the different detection modes of MALDI IMS and LC-MS/MS, MALDI IMS can detect in situ and image the distribution of the molecules [18], LC-MS/MS detects all of the molecules in the sample. We assumed that when *Ttll* knockout removes the glutamate from the tubulin C-terminal tail, the glutamate is released into the cytoplasm and then mostly stored in glutamate synaptic vesicles. With a previously reported pH of glutamatergic synaptic vesicles of approximately 5.8 [50], we supposed that the low pH of synaptic versicles influences the detection efficiency of MALDI IMS. Therefore, we detected different pH of glutamate standards and found that as the pH decreases, the glutamate signal intensity increases (Appendix A). When we detected both normal and acidic conditions, there was no difference between WT and *Ttll7* knockout mice brain sections under acidic conditions (Appendix A). We speculate there are three glutamate pools in neurons located at the microtubule C-terminal tail polyglutamylation site, cytoplasm, and synaptic vesicles. The total number of glutamates in the WT and *Ttll7* knockout mice brains is not different. When we used MALDI IMS to detect mice brains, most signals came from the lower pH synaptic versicle; thus, *Ttll* knockout mice brains showed increased glutamate concentration. Another reason is that MALDI IMS and LC-MS/MS may detect glutamate from different intracellular and extracellular sites or differentially pooled glutamates, such as polymers or monomers. 

As an important neurotransmitter, glutamates are abundantly contained, especially at the synaptic terminals [51]. About 20% of glutamate in the brain is generated from glucose [52], and some studies have suggested that glutaminases and these interacting proteins are associated with mitochondria, which are particularly abundant in axon terminals. These results indicate that glutamate is likely to be generated from glucose at the synaptic terminals [6]. However, we had not known where the rest of the glutamate comes from. Moreover, it seems to be inefficient to generate glutamate from glucose since the brain cannot store glucose, and most of the glucose in humans is stored in the liver as glycogen. This study suggests that glutamate can be stored in the brain on the tubulins as polyglutamylated chains and that TTLLs act as key regulators of glutamate in the brain.

## 5. Conclusions

In conclusion, our study demonstrates for the first time that *Ttll1* and *Ttll7* knockout altered the concentration of glutamate in the mouse brain. Tubulin polyglutamylation by these TTLLs acts as a pool of glutamate in neurons. Excess glutamate causes neurological disease. *Ttll1* and *Ttll7* can be potential therapeutic target genes for neurological diseases through the investigation of drugs modulating tubulin polyglutamylation levels.

## Figures and Tables

**Figure 1 biomolecules-13-00784-f001:**
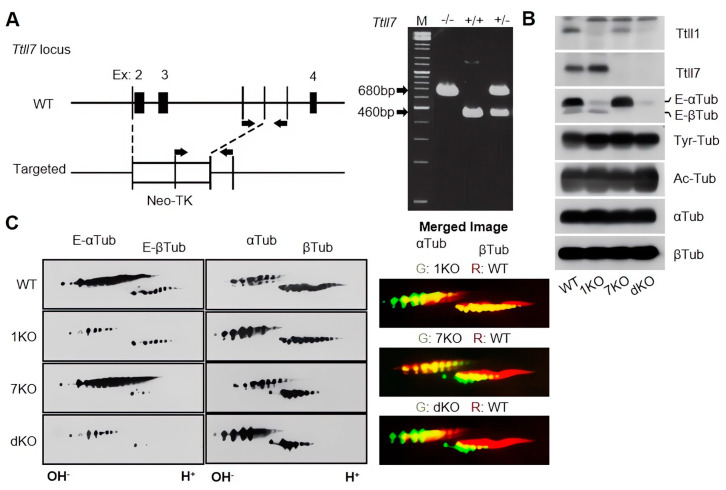
Generation of *Ttll7* knockout mice and biochemical analyses of *Ttll1* and *Ttll7* knockout mice. (**A**) Schematic representation of the generation of *Ttll7* knockout mice. Neo cassette was introduced at the *Ttll7* locus. The right panel shows PCR of the genome DNA from the transgenic mouse. M: molecular size marker, −/−: WT mouse, +/+ and +/−: homozygous and heterozygous *Ttll7* knockout mouse. (**B**) Immunoblot analysis of the brain lysates from the transgenic mice with anti-TTLL1 and TTLL7 antibodies and various anti-tubulin antibodies. (**C**) Two-dimensional electrophoresis, followed by immunoblot by using antibodies for glutamylated tubulins (left panels) or whole tubulins (center panels). Rightmost panels were generated by merging the two-dimensional electrophoresis of the WT(red) and each knockout mouse (green). Abbreviations: 1KO: *Ttll1* knockout; 7KO: *Ttll7* knockout; dKO: *Ttll1/Ttll7* double knockout; Tyr-Tub: tyrosinated tubulin; Ac-Tub: acetylated tubulin; αTub: α-tubulin; βTub: β-tubulin; E-αTub: polyglutamylated α-tubulin; E-βTub: polyglutamylated β-tubulin.

**Figure 2 biomolecules-13-00784-f002:**
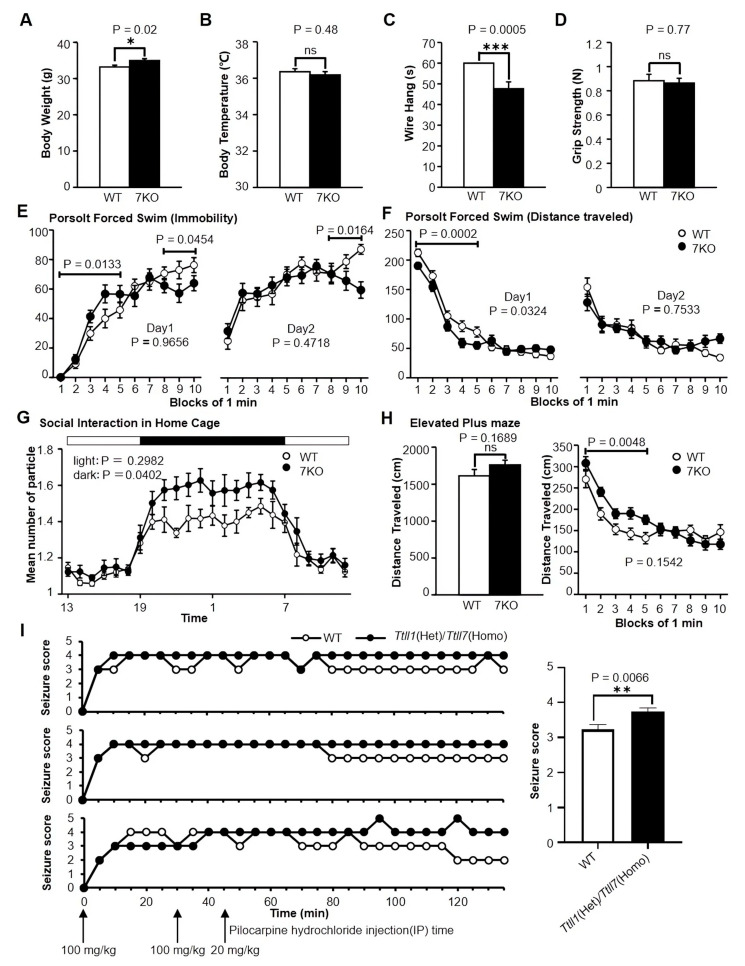
Effects of *Ttll* gene on mice behavior. (**A**) *Ttll7* knockout mice body weight was significantly heavier with *p* < 0.05, denoted by *; (**B**) the body temperature no significant difference with *p* > 0.05, denoted by ns; (**C**) the latency to fall off the wire grid was significantly shorter with *p* < 0.001, denoted by ***; (**D**) the grip strength no significant difference with *p* > 0.05, denoted by ns (WT: n = 20, 7KO: n = 20). (**E**) The immobility percentage in Porsolt forced swim test, day 1: genotype × time interaction, *p* = 0.0069; day 2: genotype × time interaction, *p* = 0.0254 (WT: n = 19, 7KO: n = 20). (**F**) The distance traveled in Porsolt forced swim test, day 1: genotype × time interaction, *p* = 0.0035 (WT: n = 19, 7KO: n = 20). (**G**) Social interaction in home cage, *Ttll7* knockout mice showed a significantly increased mean of particles during the night (*p* = 0.0402), (WT: n = 9 mice, 7KO: n = 9 mice). (**H**) There was no significant difference in distance traveled in the elevated plus maze test, but the *Ttll7* knockout mice distance traveled significantly greater in the first 5 min (genotype effect, *p* = 0.0048; genotype × time interaction, *p* = 0.0001) (WT: n = 19, 7KO: n = 20). (**I**) The kindling test was done on 3 pairs of WT and *Ttll1*(Het)/*Ttll7*(Homo) mice, *p* < 0.001, denoted by **; unpaired t test. Data represented as mean ± SEM in figure (**A**–**H**) and mean ± SD in figure (**I**). The *p* values indicate the genotype effect in one-way ANOVA (**A**–**D**, and **H** left) or two-way repeated measures ANOVA (**E**–**H** right). Abbreviations: 7KO: *Ttll7* knockout; IP: intraperitoneal injection.

**Figure 3 biomolecules-13-00784-f003:**
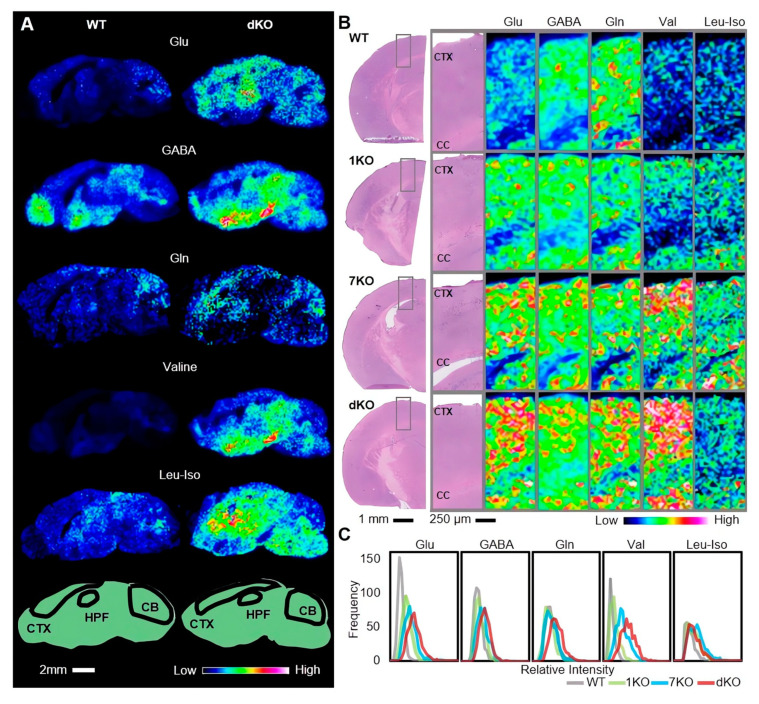
Knockout of *Ttll1* and *Ttll7* changes the distribution and concentration of glutamate in the mouse brain. (**A**) The signal intensity of glutamate, GABA, glutamine, valine, and leucine/isoleucine in the sagittal brain is shown on a rainbow color scale, CTX, HPF, and CB regions (indicated by black circles in the bellow panel), (WT: n = 3 mice; dKO: n = 3 mice). (**B**) Hematoxylin and eosin (HE) stained WT, and *Ttll* knockout mice brain coronal sections (scale bar, 1 mm). The signal intensity of glutamate, GABA, glutamine, valine, and leucine/isoleucine in the cortical area of coronal brain sections are shown in a rainbow color scale, (WT: n = 1 mouse; 1KO: n = 1 mouse; 7KO: n = 1 mouse; dKO: n = 1 mouse). (**C**) The frequency of the relative intensity of each spot in the same cortical region of *Ttll* knockout and WT mice brains is represented in line graphs. Abbreviations: Glu: glutamate; GABA: γ-aminobutyric acid; Gln: glutamine; Val: valine; Leu-Iso: leucine-isoleucine, 1KO: *Ttll1* knockout; 7KO: *Ttll7* knockout; dKO: *Ttll1/Ttll7* double knockout.

**Figure 4 biomolecules-13-00784-f004:**
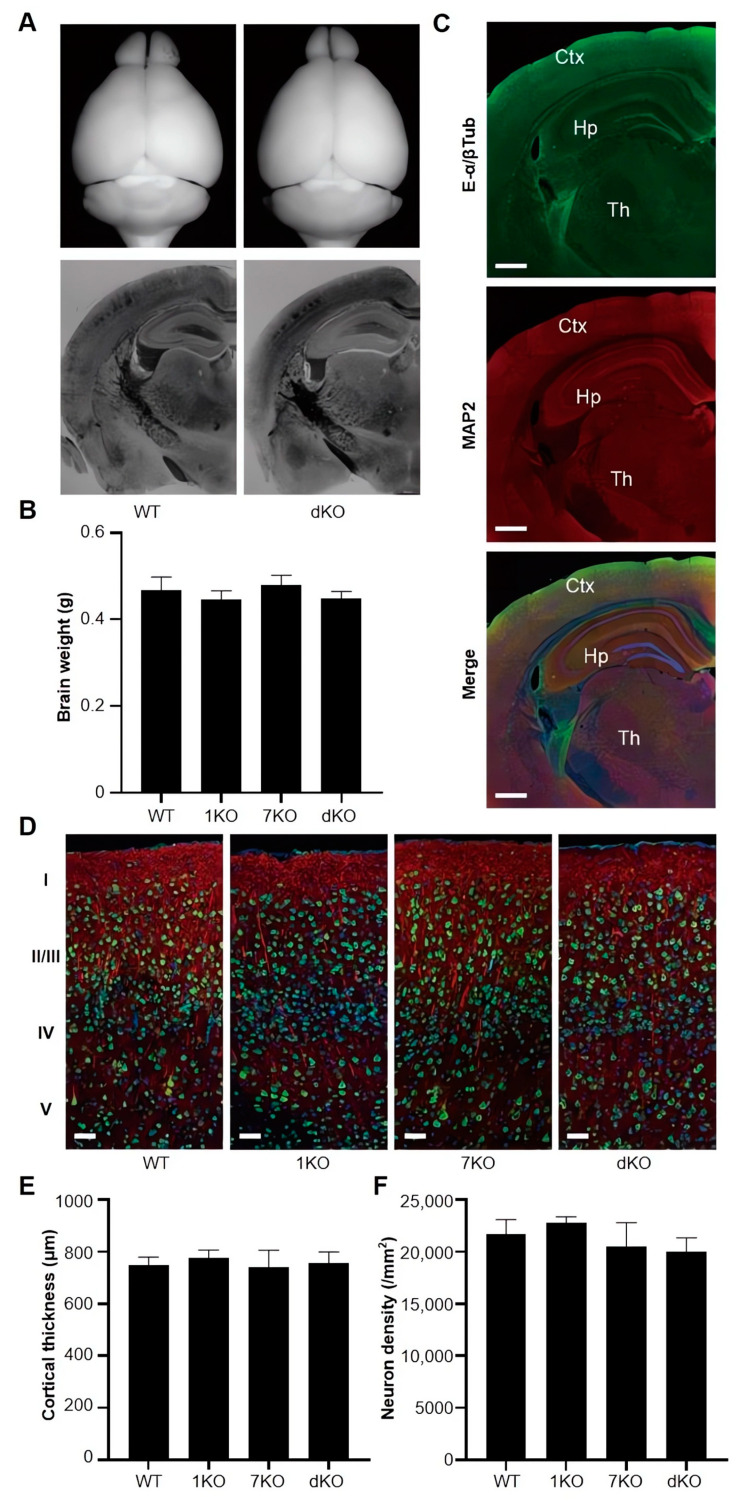
Overall structures of brains from *Ttll* knockout mice. (**A**) Whole views (upper panels) and cress sectional views (below panels) of the WT and *Ttll1/Ttll7* double knockout mice. (**B**) Brain weight of transgenic mice indicating that *Ttll* knockout did not affect the weight of the brain, (WT: n = 4 mice; 1KO: n = 4 mice; 7KO: n = 4 mice; dKO: n = 4 mice). (**C**) WT mice brain sections were reacted with antibodies against glutamylated tubulin (upper panel, green, E- α/β Tub) and MAP2 (middle panel, red). The merged Figure (below panel) shows that glutamylated tubulins are dominant on the surface of the cortex region. It was also stained by Hoechst 33342 for nucleus imaging. Scale bars: 0.5 mm. (**D**) Immunostaining of cerebral cortexes from *Ttll* knockout mice stained with NeuN antibody (green) and anti-MAP2 antibody (red). Scale bars: 20 μm. (**E**,**F**) Cortical thicknesses (**E**) and neuron densities (**F**) from *Ttll* knockout mice indicated no differences in the number of neurons in this area among WT and *Ttll* knockout mice, (WT: 9 cortical sections, n = 3 mice; 1KO: 9 cortical sections, n = 3 mice; 7KO: 9 cortical sections, n = 3 mice; dKO: 9 cortical sections, n = 3 mice). Data represented as mean ± SD in (**B**,**E**,**F**), unpaired t test. Abbreviations: 1KO: *Ttll1* knockout; 7KO: *Ttll7* knockout; dKO: *Ttll1/Ttll7* double knockout.

**Figure 5 biomolecules-13-00784-f005:**
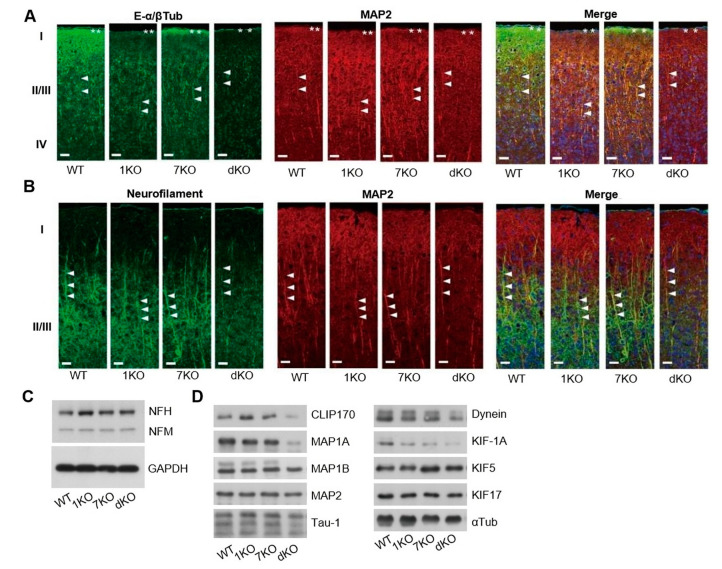
Tubulin polyglutamylation and neurofilaments in dendrites of *Ttll* knockout mice. (**A**) Cortical brain sections from *Ttll* knockout mice were stained with antibodies against glutamylated tubulin (left panels, green, E- α/β Tub), MAP2 (middle panels, red), and Hoechst 33342 for nucleus imaging (right panels, blue). (**B**) Cortical brain sections from *Ttll* knockout mice were stained with antibodies against neurofilament M (left panels, green), MAP2 (middle panels, red), and Hoechst 33342 for nucleus imaging (right panels, blue). Scale bars: 20 µm. (**C**) Immunoblot analysis of neurofilaments proteins stained by antibodies against neurofilament (NF-H and NF-M) and GAPDH as a control. (**D**) Immunoblot analysis of MAPs and motor proteins co-sedimented with brain tubulin from *Ttll* knockout mice. Abbreviations: 1KO: *Ttll1* knockout; 7KO: *Ttll7* knockout; dKO: *Ttll1/Ttll7* double knockout; E-α/βTub: polyglutamylated α/β-tubulin.

**Figure 6 biomolecules-13-00784-f006:**
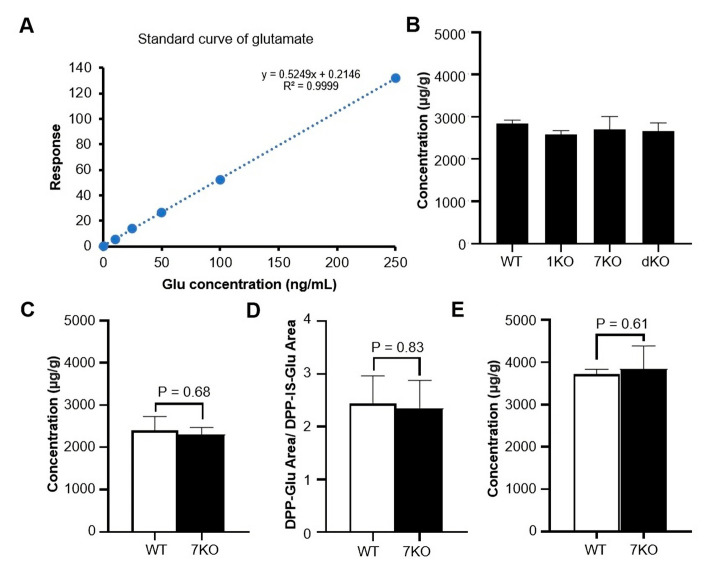
Quantitative analyses of the mice brain glutamate by LC-MS/MS. (**A**) Standard curve of glutamate in the LC-MS/MS quantitative analysis. (**B**) Quantitative analysis *Ttll* knockout mice half brain coronal sections glutamate concentration by an Acquity UPLC system with 4000 QTRAP linear ion trap quadrupole mass spectrometer. (WT: n = 3 mice; 1KO: n = 3 mice; 7KO: n = 3 mice; dKO: n = 3). (**C**) Using neutralized extraction solutions extract *Ttll7* knockout and WT mice brain glutamate, quantitative analysis by 4000 QTRAP mass spectrometer, unpaired t test. (WT: n = 3 mice; 7KO: n = 3 mice). (**D**) Quantitative analysis *Ttll7* knockout and WT mice brain DPP-Glu by 4000 QTRAP mass spectrometer, unpaired t test. (WT: n = 3 mice; 7KO: n = 3 mice). (**E**) Quantitative analysis *Ttll7* knockout and WT mice whole brain coronal sections glutamate detected by Xevo TQ-XS triple quadrupole mass spectrometer, unpaired t test. (WT: n = 6 mice; 7KO: n = 6 mice). All data represent as mean ± SD, unpaired t test. Abbreviations: 7KO: *Ttll7* knockout.

**Table 1 biomolecules-13-00784-t001:** Statistical analysis of each spot intensity of amino acids detected in the sagittal section.

	Glu		GABA		Gln		Val		Leu-Iso	
	*p*	Effect Size	*p*	Effect Size	*p*	Effect Size	*p*	Effect Size	*p*	Effect Size
CTX	<0.001	0.757	<0.001	0.332	<0.001	0.209	<0.001	0.653	<0.001	0.372
HPF	<0.001	0.761	<0.001	0.462	0.078	0.12	<0.001	0.796	<0.001	0.552
CB	<0.001	0.535	<0.001	0.431	<0.001	0.334	<0.001	0.82	<0.001	0.333

Statistical analysis of each spot intensity of amino acids detected in the sagittal section. Wilcoxon test, follow by Effect Size; 0.10–<0.3 (small effect), 0.3–<0.5 (moderate effect), and ≥0.5 (large effect). Abbreviations: Glu: glutamate; GABA: γ-aminobutyric acid; Gln: glutamine; Val: valine; Leu-Iso: leucine-isoleucine.

**Table 2 biomolecules-13-00784-t002:** Statistical analysis of each spot intensity of amino acids detected in the coronal section.

	Glu		GABA		Gln		Val		Leu-Iso	
	*p*	Effect Size	*p*	Effect Size	*p*	Effect Size	*p*	Effect Size	*p*	Effect Size
CTX	<0.001	0.572	<0.001	0.241	<0.001	0.179	<0.001	0.666	<0.001	0.172

Statistical analysis of each spot intensity of amino acids detected in the coronal section. Kruskal-Wallis test, followed by Effect Size; 0.01–<0.06 (small effect), 0.06–<0.14 (moderate effect), and ≥0.14 (large effect). Abbreviations: Glu: glutamate; GABA: γ-aminobutyric acid; Gln: glutamine; Val: valine; Leu-Iso: leucine-isoleucine.

## Data Availability

Data is contained within the article or Appendix A.

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
