# Peer review of "Tubulin Polyglutamylation by TTLL1 and TTLL7 Regulate Glutamate Concentration in the Mice Brain"

_biomolecules, 2023, doi:10.3390/biom13050784_

Round 1

Reviewer 1 Report

The authors carried out an experimental work aimed to elucidating the role of tubulin polyglutamylation mediated in the brain by tyrosine ligase-like, TTLL1 and TTLL7, which have been indicated to be important for neuronal polarity. To this aim, they generated TTLL knock-out mice and found that these genetically modified mice exhibited abnormal behaviour and an increase in the concentration of glutamate and other aminoacids.

The conclusion of these results is that tubulin polyglutamylation is useful for providing neuroinal cells with a disposable pool of glutamate.

The experimental approach used is elegant and the results are novel. Howevber, minor criticisms remain to be addressed.

Minor points

1.     The authors should highlight the translational potential of their results. In the Conclusions, they stated that TTLL-like proteins can become therapeutic targets in neurodegenerative disorders. However, given that tubulin polyglutamylation has a role for increasing the disposable pool of glutamate, and that neurodegeneration is usually triggered by excitotoxicity, they should better explain what kind of therapeutic approach could involve TTLLs.

2.     The Reference list should be updated by including more recently published papers, i.e. released in the last five years, about tubulin polyglutamylation

Reviewer 2 Report

This manuscript presents the role of TTLL1 and TTLL7 in modulation of glutamate neurotransmitter in mouse brain using a complex set of assays. The subject is of interest and the manuscript is well structured.

Comments:

The Introduction should contain more information regarding the role of glutamate neurotransmitter in healthy and diseased brain to highlight the importance of the study.  

In the M&M, Antibodies, please mention the concentration of each antibody that has been used.

In the M&M, Porsolt forced swim test, the authors mention a 10-minute test period. However, in the Results it appears that the test was performed in two days with a 5-minute test period. Please check and adjust.

In the M&M, Elevated plus maze, please explain why the authors chose to evaluate the animals for 10 minutes. The usual test period for Elevated plus maze test in 5 minutes.

In the M&M, Transmission Electron microscopy, please mention what ultrathin section means.

In the Results, lines 305-308, please remove. It is no relevant for the current manuscript.

In the Results, Ttll knockout mice showed several abnormal behaviors, it is unclear why the authors presented the information obtained only for Ttll7 knockout mice or only for Ttll1(Het)/Ttll7(Homo) mice (for kindling test).

From the Conclusions, please remove Glutamate is the most important excitatory neurotransmitter in the brain.. This fact has not been demonstrated in this study.

Reviewer 3 Report

“Tubulin polyglutamylation by TTLL1 and TTLL7 regulate glutamate concentration in the mice brain”

The manuscript presents a study evaluating how glutamate levels are regulated in the mouse brain. The authors used mouse knockout in their study. They used the LCMS method to assess the concentration of glutamate in the brain of mice. The results are well developed, a number of behavioral tests and biochemical tests have been performed.

However, in the presented work lacks information on the number of mice used in the study. In addition, the authors present the results of other neurotransmitters such as GABA, but do not provide a description of how the content of these compounds was assessed. Such information is essential for a reliable interpretation of the results.

The work is very interesting, however, the authors must pay special attention to inconsistencies and errors before publication.

Round 2
